# Novel *N*-normetazocine Derivatives with Opioid Agonist/Sigma-1 Receptor Antagonist Profile as Potential Analgesics in Inflammatory Pain

**DOI:** 10.3390/molecules27165135

**Published:** 2022-08-12

**Authors:** Rita Turnaturi, Santina Chiechio, Lorella Pasquinucci, Salvatore Spoto, Giuliana Costanzo, Maria Dichiara, Silvia Piana, Margherita Grasso, Emanuele Amata, Agostino Marrazzo, Carmela Parenti

**Affiliations:** 1Department of Drug and Health Sciences, Medicinal Chemistry Section, University of Catania, Viale A. Doria 6, 95125 Catania, Italy; 2Department of Drug and Health Sciences, Section of Pharmacology and Toxicology, University of Catania, Viale A. Doria 6, 95125 Catania, Italy; 3Oasi Research Institute—IRCCS, 94018 Troina, Italy; 4Department of Biomedical and Biotechnological Sciences, University of Catania, Via Santa Sofia, 97, 95123 Catania, Italy

**Keywords:** multitarget drugs, MOR/DOR agonists, PRE-084, formalin test

## Abstract

Although opioids and nonsteroidal anti-inflammatory drugs (NSAIDs) are the most common drugs used in persistent pain treatment; they have shown many side effects. The development of new analgesics endowed with mu opioid receptor/delta opioid receptor (MOR/DOR) activity represents a promising alternative to MOR-selective compounds. Moreover, new mechanisms, such as sigma-1 receptor (σ_1_R) antagonism, could be an opioid adjuvant strategy. The in vitro σ_1_R and σ_2_R profiles of previous synthesized MOR/DOR agonists (−)-2*R*/*S*-LP2 (**1**), (−)-2*R*-LP2 (**2**), and (−)-2*S*-LP2 (**3**) were assayed. To investigate the pivotal role of *N*-normetazocine stereochemistry, we also synthesized the (+)-2*R*/*S*-LP2 (**7**), (+)-2*R*-LP2 (**8**), and (+)-2*S*-LP2 (**9**) compounds. (−)-2*R*/*S*-LP2 (**1**), (−)-2*R*-LP2 (**2**), and (−)-2*S*-LP2 (**3**) compounds have Ki values for σ1R ranging between 112.72 and 182.81 nM, showing a multitarget opioid/σ1R profile. Instead, (+)-2*R*/*S*-LP2 (**7**), (+)-2*R*-LP2 (**8**), and (+)-2*S*-LP2 (**9**) isomers displayed a nanomolar affinity for σ1R, with significative selectivity vs. σ2R and opioid receptors. All isomers were evaluated using an in vivo formalin test. (−)-2*S*-LP2, at 0.7 mg/kg i.p., showed a significative and naloxone-reversed analgesic effect. The σ1R selective compound (+)-2*R*/*S*-LP2 (**7**), at 5.0 mg/kg i.p., decreased the second phase of the formalin test, showing an antagonist σ1R profile. The multitarget or single target profile of assayed *N*-normetazocine derivatives could represent a promising pharmacological strategy to enhance opioid potency and/or increase the safety margin.

## 1. Introduction

Inflammatory pain is a pathological condition related to an increased sensitivity caused by tissue damage. The increasing incidence and impact of inflammatory diseases have encouraged the search for new pharmacological strategies to face them. The conventional therapies for inflammation, including steroidal (SAID), nonsteroidal anti-inflammatory drugs (NSAID), and disease-modifying antirheumatic drugs (DMARDs) [1,2], have shown many side effects and deficiencies. Moreover, opioid therapy for inflammatory pain for the treatment of chronic diseases is often associated with several side effects. Therefore, to date, the treatment of chronic inflammatory pain is one of the unmet clinical needs for patients.

Recently, the delta opioid receptor (DOR) has become an attractive target as it is characterized by limited side effects compared to mu opioid receptor (MOR) agonists [3]. Moreover, simultaneous MOR-DOR activation, supported by co-expression of both receptors in areas involved in pain modulation [4], has been indicated as a strategic mechanism for chronic pain treatment [5]. In fact, MOR-DOR agonists showed improved antinociception and a low propensity to develop side effects [5,6,7]. Favorable pharmacological interactions were also demonstrated for our previously synthesized MOR/DOR agonist (−)-2*R*/*S*-LP2 (**1**) (Figure 1) in inflammatory and neuropathic behavioural signs in mice and rats [8,9,10,11]. Furthermore, new targets involved in inflammatory pain modulation have also been studied, and among these, sigma-1 receptor (σ_1_R) showed a determining role in pain transmission [12].

σ_1_R is a 24 kDa protein of 223 aminoacids anchored to the endoplasmic reticulum and plasma membranes that acts as a molecular chaperone, modulating the functionality of different receptors and ion channels [13,14]. It is expressed in peripheral organs and many areas of the central nervous system (CNS), including key areas for pain modulation, such as dorsal root ganglia, superficial layers of dorsal horn, locus coeruleus, and periaqueductal grey matter [15]. Literature data demonstrate that most of the endogenous ligands for σ_1_R are steroids synthesized in the nervous system (neurosteroids), such as pregnenolone-sulfate (PREG-S) and dehydroepiandrosterone sulfate (DHEA-S), that have a role as σ_1_R agonists [16]. On the other hand, progesterone is an endogenous antagonist of this receptor, and it induces down-regulation of TRPV1 expression in the plasma membrane of sensory neurons, causing a decrease in capsaicin-induced nociceptive responses [17].

The σ1R modulation of analgesia was first proposed by Chien and Pasternak, who identified σ1R as a potent anti-opioid endogenous system [18]. It has been demonstrated that σ1R agonists are able to strongly counteract spinal and supraspinal opioid-induced analgesia, while σ1R antagonists potentiate opioid-induced analgesia [19]. Furthermore, selective σ1R antagonists have demonstrated analgesic efficacy in acute and chronic inflammatory pain models [20,21].

Several hypotheses explain the involvement of σ1R in inflammatory pain modulation. One possible mechanism is that σ1R activation enhances the bradykinin induced Ca^2+^ release and nitric oxide (NO) signalling, thereby strengthening the inflammatory process [22,23]. Moreover, a peripheral σ1R-related mechanism, proposed by Tejada et al. [24], might be even more important in the modulation of inflammatory pain sustained by a conspicuous enhancement of peripheral sensitization [25]. Additionally, pain sensitization after peripheral inflammation involves plastic changes characterized by an increase in spinal excitatory neurotransmission [26]. It has been described as a modulatory role of σ1Rs in spinal sensitization and point to *N*-methyl-*D*-aspartate (NMDA) receptors and Ca^2+^-dependent intracellular cascades as underlying mechanisms. The overall effect of activating σ1Rs is to increase intracellular Ca^2+^ concentration by potentiating Ca^2+^ entry at the plasma membrane level (NMDA-induced Ca^2+^ influx) and Ca^2+^ mobilization from endoplasmic stores (IP3-induced Ca^2+^ mobilization) [27]. The modulatory role of σ1R in pain transmission was identified using σ1R knockout (KO) mice that showed attenuated pain responses in both phases of the formalin test [28,29], confirmed also by the administration of σ1R antagonists [24].

Based on these findings, the aim of our work was to evaluate the opioid and/or σ1R functional profile of *N*-normetazocine-based ligands as potential therapeutic agents for inflammatory pain management. Previously, we synthesized the (−)-2*R*-LP2 (**2**) and (−)-2*S*-LP2 (**3**) isomers (Figure 1) characterized by a peculiar in vitro and in vivo profile [30]. In the present work, to investigate the pivotal role of *N*-normetazocine stereochemistry in their pharmacological fingerprint, we also synthesized the (+)-2*R*/*S*-LP2 (**7**), (+)-2*R*-LP2 (**8**), and (+)-2*S*-LP2 (**9**) compounds (Figure 1).

First, by competition binding assays, we evaluated the affinity for opioid receptors and σ1R, σ2R of all (−)-LP2 and (+)-LP2 isomers. Moreover, based on the role of simultaneous MOR-DOR activation and σ1R antagonism in inflammatory pain modulation, we tested all (−)-LP2 and (+)-LP2 isomers in the mouse formalin test. To highlight the contribution of opioid receptors and σ1R in the observed effects, coadministration of the opioid antagonist naloxone or selective σ1R agonist PRE-084, respectively, were evaluated.

## 2. Results and Discussion

### 2.1. Chemistry

The resolution of (±)-*cis*-*N*-normetazocine and the synthesis of intermediates **4**, **5**, and **6** were carried out as previously reported [8,31] (See Appendix A). Compounds (+)-*2R/S*-LP2 (**7**), (+)-2*R*-LP2 (**8**), and (+)-2*S*-LP2 (**9**) were obtained by alkylation of (+)-*cis*-*N*-normetazocine with the respective tosylated alcohols **4**–**6** (Figure 1). All final compounds were characterized by ^1^H and ^13^C NMR (Appendix A) and elemental analysis (Appendix A).

### 2.2. Radioligand Binding Assays

The σ1R and σ2R affinity of all (−)-LP2 and (+)-LP2 isomers were evaluated by competition binding assays using [^3^H]-(+)-pentazocine ([^3^H]-PTZ) and [^3^H]-DTG, respectively. Moreover, newly synthesized (+)-LP2 isomers were evaluated vs. MOR, DOR, and kappa opioid receptor (KOR) with the respective [^3^H]-DAMGO, [^3^H]-Delthorphin, and [^3^H]-U69,593 radioprobe (Table 1). Haloperidol, (+)-pentazocine, DTG, BD-1063, DAMGO, naltrindole, and (−)-U50,488 are reported as internal controls. A similarity test, by calculating the Tanimoto structural similarity index (Tsc) using the online software, https://chemminetools.ucr.edu/similarity/ accessed on 1 August 2022, was conducted on all tested compounds (Appendix A).

(−)-2*R*/*S*-LP2 (**1**), (−)-2*R*-LP2 (**2**), and (−)-2*S*-LP2 (**3**), with a previously demonstrated MOR/DOR agonist profile, have Ki values for σ1R ranging between 112.72 and 182.81 nM. Thus, in levorotatory isomers, dual opioid/σ1R affinities were displayed. Furthermore, for (−)-2*R*/*S*-LP2 (**1**) and (−)-2*R*-LP2 (**2**) nanomolar Ki values vs. σ2R were revealed. In addition, all (+)-2*R*/*S*-LP2 (**7**), (+)-2*R*-LP2 (**8**), and (+)-2*S*-LP2 (**9**) compounds displayed a double-digit nanomolar affinity for σ1R, with significative selectivity vs. σ2R and opioid receptors. Indeed, contrarily to levo isomers, whose Ki σ2R/σ1R ratios ranged from 0.38 to 9, dextro isomers σ2R/σ1R selectivity ratios were 87, 40 and 47, for (+)-2*R*/*S*-LP2 (**7**), (+)-2*R*-LP2 (**8**), and (+)-2*S*-LP2 (**9**), respectively. Moreover, opioid/σ1R selectivity ratios in all dextro isomers ranged from 16.5 to 53.6. Thus, an affinity profile of selective σ1R ligands was shown.

The *N*-normetazocine nucleus has mainly been employed for the design of opioid analgesics, although it is a versatile structure. In fact, its stereochemistry plays a crucial role in directing *N*-normetazocine-based compounds to different targets [30]. The literature data reports that the (−)-(1*R*,5*R*,9*R*) configuration mainly interacts with opioid receptors [32], while, the (+)-(1*S*,5*S*,9*S*) antipode is able to bind σ1R. Alazocine ((+)-SKF-10,047) was the first compound discovered that highlighted a remarkable σ1R affinity. Conversely its (−)-isomers are more tightly bound to MOR and KOR [33]. Furthermore, Carrol et al. [34] synthesized (+)- and (−)-*N*-substituted analogues of (+)-SKF-10,047, proving that σ1R affinity and selectivity of all compounds were influenced by the nature of the *N*-substituent and the *N*-normetazocine stereochemistry. Subsequently, regardless of their stereochemistry, the (+)- and (−)-phenazocines, characterized by an *N*-2-phenylethyl substituent, showed opioid agonist/σ1R antagonist profiles and a significant in vivo analgesia [35]. Differently, the analgesic effect of (+)-LP1, with an *N*-phenylpropanamide substituent, was not reversed by naloxone, suggesting a σ1R antagonist profile [36]. Thus, the stereochemistry of the *N*-normetazocine scaffold as well as the size, electronic, and steric properties of its *N*-substituents could shift the functional profile from selective σ1R to mixed-target compounds.

Our new data confirmed this variability in targeting opioid receptors and/or σ1R. All our levo isomers, (−)-2*R*/*S*-LP2 (**1**), (−)-2*R*-LP2 (**2**), and (−)-2*S*-LP2 (**3**), showed a multitarget profile, while the new synthesized dextro isomers, (+)-2*R*/*S*-LP2 (**7**), (+)-2*R*-LP2 (**8**), and (+)-2*S*-LP2 (**9**), are more stringently linked at σ1R profile. Thus, *N*-normetazocine stereochemistry and *N*-substituent moiety seem to modify the affinity profile.

### 2.3. Formalin Test

Mice injected with 5% formalin into the mid-plantar surface of the right hind paw developed the characteristic formalin-induced pain behaviour, evidenced by a biphasic flinching and lifting/licking response. The time course of formalin-induced behaviours evidenced an immediate and acute nociceptive activity (0–5 min, phase I) that, after a short quiescent period (5–10 min, interphase), was followed by a more prolonged response (10–60 min, phase II).

To evaluate the systemic antinociceptive effect in both phases of the formalin test, vehicle, (−)-2*R*-LP2 (**2**) (2mg/kg), (−)-2*S*-LP2 (**3**) (0.7 mg/kg), (+)-2*R*/*S*-LP2 (**7**) (5 mg/kg), and (+)-2*R*-LP2 (**8**) (5 mg/kg) were administrated intraperitoneally (i.p.) 20 min before formalin injection, at doses chosen in accordance to previous studies [28]. As shown in Figure 2, the administration of compound (−)-2*S*-LP2 showed a significant analgesic effect, reducing the flinching/licking time in both phases of the formalin test. The (−)-2*R*-LP2 isomer showed the same profile with a minor potency, proving that the stereocenter influences pharmacological characteristics such as affinity, as demonstrated by radioligand binding assays.

To prove that the systemic antinociceptive effect was mainly mediated by the opioid system, we administrated a subcutaneous (s.c.) solution of naloxone hydrochloride (3 mg/kg), a non-selective opioid receptor antagonist, 20 min before the injection of (−)-2*S*-LP2 (the more representative of the two compounds under study). As shown in Figure 3, naloxone antagonized the antinociceptive effect of (−)-2*S*-LP2, confirming its main effect through the opioid system. Moreover, given the affinity for σ1R of the compounds, a contribution of this receptor to the opioid-mediated antinociception effect should be considered.

It is well established that σ1R is involved in nociception, and the effects reported with σ1R ligands are consistent with a role of this receptor in pain hypersensitivity [37]. Thus, considering the results obtained in the radioligand binding assay with a (+)-2*R*/*S*-LP2 K_i_ value two times lower than (+)-2*R*-LP2 and (+)-2*S*-LP2 and with a significant σ2R/σ1R and opioid receptors/σ1R selectivity ratio, we then moved to test the analgesic efficacy of compounds **7**, **8**, and **9** in the mouse formalin test.

As shown in Figure 4, administration of (+)-2*R/S*-LP2 (**7**) at a dose of 5 mg/kg 20 min before formalin injection by the i.p. route did not alter the severity of the first phase of pain. The second phase, instead, was significantly decreased, as evidenced by a reduction in both the number of flinches and the lifting/licking time as compared with vehicle-treated mice. (+)-2*R*-LP2 (**8**) effect was not significative. The same result was obtained with the isomer (+)-2*S*-LP2 (**9**).

In inflammatory pain, spinal cord neuronal sensitization occurs during the second phase of the formalin test. As NMDA receptor activation plays an important role in formalin-induced pain the obtained results apparently support the hypothesis that σ1R facilitates the expression of formalin-induced pain, through their known ability to modulate NMDA-mediated responses [38].

To verify the antagonist profile of (+)-2*R/S*-LP2 (**7**) vs. σ1R, a solution of PRE-084 hydrochloride (32 mg/kg), a known high-selective σ1R agonist, was administrated s.c. 20 min before the (+)-2*R/S*-LP2 (**7**) injection. Injection of PRE-084 hydrochloride alone did not change the severity of the first and second phases of formalin-induced pain [36,39]. Prior microinjection of PRE-084 hydrochloride (32 mg/kg s.c.) prevented (+)-2*R/S*-LP2 (**7**) antinociceptive effects, asserting its antagonist profile vs. σ1R (Figure 5).

## 3. Materials and Methods

### 3.1. General Remarks

All commercial chemicals were purchased from Merck (Darmstadt, Germany) and were used without further purification. (±)-*cis*-*N*-normetazocine was obtained from Fabbrica Italiana Sintetici. Melting points were determined in open capillary tubes with a Büchi 530 apparatus and are uncorrected. Analytical TLC was performed on silica gel 60 F_254_ aluminium sheets (Merck) with a fluorescent indicator. Components were visualized by UV light (λ = 254 nm) and iodine vapour. Flash column chromatography was carried out on Merck silica gel 60 (230–400 mesh). Optical rotations were determined in MeOH solution with a Perkin-Elmer 241 polarimeter. ^1^H and ^13^C NMR spectra were routinely recorded on a Varian Inova-200 spectrometer in CDCl_3_ solution; chemical shifts δ are expressed in ppm with reference to tetramethylsilane as an internal standard. Elemental analyses (C, H, and N) were performed on a Carlo Erba 1106 analyzer and the analysis results were within ± 0.4% of the theoretical values. All reported compounds had a purity of at least 95%.

### 3.2. General Procedure for the Synthesis of Compounds ***7***−***9***

A mixture of (+)-*cis*-*N*-normetazocine (1.63 mmol, 1 eq), the appropriate tosylate intermediates (1.63 mmol, 1 eq), NaHCO_3_ (2.45 mmol, 1,5 eq), and a catalytic amount of KI was stirred in DMF at 65 °C for 24 h. After cooling, the reaction mixture was filtered and concentrated under vacuum to remove DMF. The resulting residue was purified by flash chromatography on a silica gel column using CH_2_Cl_2_/C_2_H_5_OH (97:3 *v*/*v*) as an eluent to yield the final compounds (**7**–**9**). The free bases were converted into hydrochloride salts by dissolution in a minimum amount of THF and adding 1N HCl in Et_2_O to the solution.


(2S,6S,11S)-3-(2-methoxy-2-phenylethyl)-6,11-dimethyl-1,2,3,4,5,6-hexahydro-2,6-methanobenzo[d]azocin-8-ol (**7**)


Brown solid (35%). Mp 176–180 °C. ${\left[\alpha \right]}_{D}^{25}$ = +75.7° (c 1.004, MeOH). TLC CH_2_Cl_2_/C_2_H_5_OH (95:5 *v*/*v*) Rf = 0.38. ^1^H NMR (200 MHz, CDCl_3_, free base) δ 7.19–7.16 (m, 5H, CH aryl), 6.86 (d, 1H, *J* = 8.0 Hz, CH benzomorphan), 6.685 (d, 1H, *J* = 2.0 Hz, CH benzomorphan), 6.61 (dd, 1H, *J* = 10.0, 2.0 Hz, CH benzomorphan), 4.65 (d, 1H, CH-O), 3.14 (s, 3H, -OCH_3_), 2.92–2.68 (m, 6H, CH, CH_2_ benzomorphan and CH_2_), 2.33–1.97 (m, 3H, CH benzomorphan), 1.26 (s, 3H, CH_3_ benzomorphan), 1.18 (m, 1H, CH benzomorphan), 0.73 (d, 3H, CH_3_ benzomorphan). ^13^C NMR (50 MHz, CDCl_3_, free base) δ 154.79, 139,79, 136.20, 127.04, 126.27, 125.34, 122.93, 122.66, 113.08, 111.31, 82.02, 60.75, 60.40, 58.58, 46.48, 38.09, 35.52, 35.16, 23.09, 21.59, 11.77 (Appendix A). Anal (C_23_H_29_NO_2_·HCl) C, H, N (Appendix A).


(2S,6S,11S)-3-((*R*)-2-methoxy-2-phenylethyl)-6,11-dimethyl-1,2,3,4,5,6-hexahydro-2,6-methanobenzo[d]azocin-8-ol (**8**)


Brown solid (40%). Mp 177–180 °C. ${\left[\alpha \right]}_{D}^{25}$ = +31.4° (c 1.018, MeOH). TLC CH_2_Cl_2_/C_2_H_5_OH (95:5 *v*/*v*) Rf = 0.38.^1^H NMR (200 MHz, CDCl_3_, free base) δ 7.26–7.19 (m, 5H, CH aryl), 6.83 (d, 1H, *J* = 8.0 Hz, CH benzomorphan), 6.64 (d, 1H, *J* = 2.0 Hz, CH benzomorphan), 6.55 (dd, 1H, *J* = 8.0, 2.0 Hz, CH benzomorphan), 4.36 (d, 1H, CH-O), 3.13 (s, 3H, -OCH_3_), 2.85–2.55 (m, 6H, CH, CH_2_ benzomorphan and CH_2_), 2.23–2.17 (m, 1H, CH benzomorphan), 1.88–1.82 (m, 2H, CH_2_ benzomorphan), 1.24 (s, 3H, CH_3_ benzomorphan), 1.19 (m, 1H, CH benzomorphan), 0.735 (d, 3H, CH_3_ benzomorphan). ^13^C NMR (50 MHz, CDCl_3_, free base) δ 155.06, 148.96, 147.07, 129.28, 128.11, 126.23, 122.47, 121.29, 113.69, 112.99, 84.54, 67.08, 56.72, 47.02, 46.23, 35.97, 33.1.15.27, 31.82, 29.15, 25.02, 13.66 (Appendix A). Anal (C_23_H_29_NO_2_·HCl) C, H, N (Appendix A).


(2*S*,6*S*,11*S*)-3-((*S*)-2-methoxy-2-phenylethyl)-6,11-dimethyl-1,2,3,4,5,6-hexahydro-2,6-methanobenzo[d]azocin-8-ol (**9**)


Brown solid (33%). Mp 177–180 °C; ${\left[\alpha \right]}_{D}^{25}$ = +44.0° (c 0.50, MeOH). TLC CH_2_Cl_2_/C_2_H_5_OH (95:5 *v*/*v*) Rf = 0.40. ^1^H NMR (200 MHz, CDCl_3_, free base) δ 7.20–7.19 (m, 5H, CH aryl), 6.83 (d, 1H, *J* = 8.0 Hz, CH benzomorphan), 6.65 (d, 1H, *J* = 2.0 Hz, CH benzomorphan), 6.55 (dd, 1H, *J* = 8.0, 2.0 Hz, CH benzomorphan), 4.44 (d, 1H, CH-O), 3.15 (s, 3H, -OCH_3_), 2.93–2.60 (m, 6H, CH, CH_2_ benzomorphan and CH_2_), 2.19–1.85 (m, 3H, CH and CH_2_ benzomorphan), 1.25 (s, 3H, CH_3_ benzomorphan), 1.18 (m, 1H, CH benzomorphan), 0.755 (m, 3H, CH_3_ benzomorphan). ^13^C NMR (50 MHz, CDCl_3_, free base) δ 153.96, 142.62, 140.42, 129.45, 128.00, 127.61, 126.57, 125.69, 113.72, 112.25, 81.52, 61.38, 58.12, 57.00, 46.02, 40.90, 39.84, 36.13, 24.39, 22.97, 14.59 (Appendix A). Anal (C_23_H_29_NO_2_·HCl) C, H, N (Appendix A).

### 3.3. Radioligand Binding Assays

#### Animals

Brain and liver homogenates for σ1R and σ2R binding assays were prepared from male Dunkin-Hartley guinea pigs and Sprague-Dawley rats, respectively (ENVIGO RMS S.R.L., Udine, Italy). Animals (200–250 g) were euthanized with CO_2_ in a euthanasia chamber and sacrificed by decapitation. Guinea pig brains without cerebellum (~2.5 g each) and rat livers (~7 g each) were kept on dry ice and stored at −80 °C. Brains for MOR, DOR, and KOR binding assays were explanted from male Sprague-Dawley rats and Dunkin-Hartley guinea pigs (Italian Minister of Health project code 335/1984F.N.JLT; ENVIGO RMS S.R.L., Udine, Italy). Animals (200–250 g guinea pigs) were euthanized with CO_2_ in a euthanasia chamber and sacrificed by decapitation.

### 3.4. Radioligand Binding Assays for σ1R and σ2R

#### 3.4.1. Materials

[^3^H] (+)-Pentazocine (26.9 Ci/mmol) and [^3^H]1,3-di-o-tolylguanidine ([^3^H]DTG, 35.5 Ci/mmol) were purchased from PerkinElmer (Zaventem, Belgium). Ultima Gold MV Scintillation cocktail was from PerkinElmer (Milan, Italy). All the other materials were obtained from Merck Life Science S.r.l. (Milan, Italy). The test compound solutions were prepared by dissolving approximately 10 µmol of the test compound in DMSO so that a 10 mM stock solution was obtained. The required test concentrations for the assay (from 10^−5^ to 10^−11^ M) have been prepared by diluting the DMSO stock solution with the respective assay buffer. All experiments were performed using ultrapure water obtained with a Millipore Milli-Q Reference Ultrapure Water Purification System. All the laboratory glassware was first washed with a 6 M HCl water solution and then rinsed with ultrapure water.

#### 3.4.2. Preparation of Membrane Homogenates from Pig Brain

Fresh guinea pig brain cortices (~25 g) were homogenized in two portions with 10 volumes of ice-cold Tris (50 mM, pH 7.4) containing 0.32 M sucrose with a Potter-Elvehjem glass homogenizer. The suspension was centrifuged at 1030× *g* for 10 min at 4 °C. The supernatant was separated and centrifuged at 41,200× *g* for 20 min at 4 °C. The obtained pellet was suspended with 3 volumes of ice-cold Tris (50 mM, pH 7.4), incubated at RT for 15 min, and centrifuged at 41,200× *g* for 15 min at 4 °C. The final pellet was resuspended with ~2 volumes of ice-cold Tris buffer, and frozen at −80 °C in ~1 mL portions containing about 5 mg protein/mL [40,41].

#### 3.4.3. Preparation of Membrane Homogenates from Rat Liver

Rat livers (~21 g) were cut into small pieces with a scalpel and homogenized in two portions with 6 volumes of cold 0.32 M sucrose with a Potter-Elvehjem glass homogenizer. The suspension was centrifuged at 1030× *g* for 10 min at 4 °C. The supernatant was separated and centrifuged at 31,100× *g* for 20 min at 4 °C. The pellet was resuspended with 6 volumes of ice-cold Tris buffer (50 mM, pH 8) and incubated at rt for 30 min. Then, the suspension was centrifuged at 31,100× *g* for 20 min at 4 °C. The final pellet was resuspended with 6 volumes of ice-cold Tris buffer and stored at −80 °C in ~1 mL portions containing about 6 mg of protein/mL [40,41].

#### 3.4.4. Protein Determination

The protein concentration was determined by Bradford’s method. The Bradford solution was prepared by dissolving 10 mg of Coomassie Brilliant Blue G 250 in 5 mL of 95% ethanol. To this solution, 10 mL of 85% phosphoric acid were added, and the mixture was stirred and filled to a total volume of 100 mL with ultrapure water. The calibration was carried out with bovine serum albumin as a standard at different concentrations. In a 96-well plate, 30 µL of the calibration solution or 30 µL of the membrane receptor preparation were mixed with 240 µL of the Bradford solution, respectively. After 5 min of incubation at rt, the UV absorbance was measured at λ = 595 nm using a microplate spectrophotometer reader (Synergy HT, BioTek).

#### 3.4.5. σ1. R Ligand Binding Assays

In vitro σ_1_R ligand binding assays were carried out in a Tris buffer (50 mM, pH 7.4) for 150 min at 37 °C. The thawed membrane preparation of guinea pig brain cortex (250 μg/sample) was incubated with increasing concentrations of test compounds and [^3^H] (+)-pentazocine (2 nM) in a final volume of 0.5 mL. The *K*_d_ value of [^3^H] (+)-pentazocine was 2.9 nM. Unlabelled (+)-pentazocine (10 μM) was used to measure non-specific binding. Bound and free radioligand were separated by fast filtration under reduced pressure using a Millipore filter apparatus through Whatman GF/6 glass fiber filters, which were pre-soaked in a 0.5% poly (ethyleneimine) water solution for 120 min. Each filter paper was rinsed three times with 3 mL of ice-cold Tris buffer (50 mM, pH 7.4), dried at rt, and incubated overnight with 3 mL of scintillation cocktail into pony vials. The bound radioactivity has been determined using a liquid scintillation counter (Beckman LS 6500).

#### 3.4.6. σ2. R Ligand Binding Assays

In vitro σ_2_R ligand binding assays were carried out in Tris buffer (50 mM, pH 8.0) for 120 min at rt. The thawed membrane preparation of rat liver (250 μg/sample) was incubated with increasing concentrations of test compounds and [^3^H]DTG (2 nM) in the presence of (+)-pentazocine (5 µM) as σ_1_R masking agent in a final volume of 0.5 mL. The *K*_d_ value of [^3^H]DTG was 17.9 nM. Non-specific binding was evaluated with unlabeled DTG (10 μM). Bound and free radioligand were separated by fast filtration under reduced pressure using a Millipore filter apparatus through Whatman GF/6 glass fiber filters, which were presoaked in a 0.5% poly(ethyleneimine) water solution for 120 min. Each filter paper was rinsed three times with 3 mL of ice-cold Tris buffer (10 mM, pH 8), dried at rt, and incubated overnight with 3 mL of scintillation cocktail into pony vials. The bound radioactivity has been determined using a liquid scintillation counter (Beckman LS 6500).

#### 3.4.7. Data Analysis

The *K*_i_-values were calculated with the program GraphPad Prism^®^ 5.0 (GraphPad Software, San Diego, CA, USA). The *K*_i_-values are given as the mean value ± SD from at least two independent experiments performed in duplicate.

### 3.5. Radioligand Binding Assays for Opioid Receptors

#### 3.5.1. Materials

[^3^H]-DAMGO (48.4 Ci/mmol), [^3^H]-(2-D-Ala)-[Tyrosyl-3,5-] DELTORPHIN II (54.7 Ci/mmol), and [^3^H]-U69,593 (49.3 Ci/mmol) were purchased from PerkinElmer (Zaventem, Belgium). Unlabeled naloxone hydrochloride, DAMGO, (–)-U50,488, and Naltrindole hydrochloride were purchased from Sigma-Aldrich (St. Louis, MO, USA). The Ultima Gold MV Scintillation cocktail was from PerkinElmer (Milano, Italy). A 10 mM stock solution was obtained by dissolving the test compound in DMSO and then diluting with the assay buffer to obtain the required test concentrations for the assay (from 10^−5^ to 10^−9^ M). All experiments were performed using ultrapure water obtained with a Millipore Milli-Q Reference Ultrapure Water Purification System (Millipore, Burlington, MA, USA). The bound radioactivity has been determined using a Beckman LS 6500 liquid scintillation counter (Beckman Coulter, Brea, CA, USA).

#### 3.5.2. Preparation of Membrane Homogenates from Sprague-Dawley Rat Brains for MOR and DOR Binding Assays or Guinea Pig Brains for KOR Binding Assays

Sprague-Dawley rat brains (for MOR and DOR binding assays) or guinea pig brains (for KOR binding assays) were homogenized in ice-cold Tris buffer (50 mM, pH 7.4) by using a Dounce glass homogenizer (Wheaton, Millville, NJ, USA) with a loose inner tolerance pestle first and a tight inner tolerance pestle later in a cylindrical glass tube of 40 mL volume. The suspension was centrifuged at 40,000× *g* for 20 min at 4 °C (Beckmann J2–20 centrifuge and a JA-21 rotor). The pellet was resuspended in ice-cold Tris buffer and then incubated at 37 °C for 30 min to remove endogenous ligands. After incubation, the suspension was centrifuged at 40,000× *g* for 20 min at 4 °C and the final pellet was resuspended in ice-cold Tris buffer and frozen at −80 °C in ~1 mL portions containing about 10 mg protein/mL.

#### 3.5.3. Protein Determination

The protein concentration was determined by the Bradford method. The calibration curve was built with bovine serum albumin as the standard compound at 7 different concentrations, ranging from 60 µg/mL to 210 µg/mL with blank correction. In a 96-well plate, 30 µL of the calibration solution or 30 µL of the membrane receptor preparation were mixed with 240 µL of the Bradford solution (10 mg of Coomassie Brilliant Blue G 250 in 5 mL of 95% ethanol, 10 mL of 85% phosphoric acid, and water up to 100 mL; with ultrapure water). After 5 min of incubation at RT, the UV absorbance was measured at λ = 595 nm using a microplate spectrophotometer reader (Synergy HT, BioTek, Winooski, VT, USA).

#### 3.5.4. Opioid Receptor Ligand Binding Assays

MOR and DOR binding experiments were carried out by incubating 400 μg/sample and 500 μg/sample of rat brain membranes, respectively, for 45 min at 35 °C with 1 nM [^3^H]-DAMGO or 2 nM [^3^H]-Deltorphin II (2-D-Ala)-[Tyrosyl-3,5-^3^H] in 50 mM Tris-HCl (pH 7.4). For KOR binding assays, guinea pig brain membranes (400 μg/sample) were incubated for 30 min at 30 °C with 1 nM [^3^H]-U69,593. Test compounds were added to a final volume of 1 mL. The *K*_d_ values of [^3^H]DAMGO, [^3^H]-Deltorphin II (2-D-Ala)-[Tyrosyl-3,5-^3^H]and [^3^H]-U69,593 were 1.0, 1.5, and 2.3 nM, respectively. Nonspecific binding was assessed in the presence of 10 μM unlabelled naloxone. The reaction was terminated by filtering the solution under reduced pressure using a Millipore filter apparatus through Whatman glass fiber filters, GF/C for MOR and DOR and GF/B for KOR, presoaked for 1h in a 0.1% poly(ethyleneimine) solution. Filters were washed with 50 mM ice-cold Tris-HCl buffer (2 × 4 mL), dried at rt, soaked overnight in 4 mL of scintillation cocktail into 6 mL pony vials and counted on a liquid scintillation counter.

#### 3.5.5. Data Analysis

Ki values were calculated using nonlinear regression analysis to fit a logistic equation to the competition data using GraphPad Prism 6.0 (GraphPad Software Inc., San Diego, CA, USA).

### 3.6. In Vivo Pharmacology

#### 3.6.1. Animals

Experiments were performed on male CD1 mice (Envigo Laboratories, Indianapolis, IN, USA) aged between 8 and 12 weeks. Mice were housed in 5 animals per cage under a 12/12 h light/dark cycle at a constant temperature (23–25 °C) with free access to food and water, and were allowed to acclimate for at least one week upon arrival before starting all experiments, which were conducted between 9:00 a.m. and 3:00 p.m. This study was executed according to the European Communities Council directive and Italian regulation (EEC Council 2010/63/EU and Italian D.Lgs. no. 26/2014) to replace, reduce, and refine the use of laboratory animals. All procedures were approved by the ethical committee of the University of Catania (OPBA) and by the Italian Ministry of Health (authorization n° 385/2021-PR).

#### 3.6.2. Behavioral Experiment

Before starting every experiment, mice were randomly assigned to each experimental group and allowed to acclimate in the room for 20 min. The results were analyzed by a researcher blind to the treatment procedure.

#### 3.6.3. Drugs

Formalin and naloxone hydrochloride were purchased from Merck. PRE-84 synthesis was reported in the Appendix A. All compounds were dissolved in pyrogen-free isotonic saline and dimethyl sulfoxide, (DMSO 0.5% *v*/*v*). All LP compounds were administrated i.p. 20 min before formalin injection. Naloxone hydrochloride s.c. and PRE-084 hydrochloride s.c. were administered 20 min before the injection of the LP compounds.

#### 3.6.4. Mouse Formalin Test

Formalin solution (5%, 10 μL) was administered subcutaneously into the plantar surface of the right hind paw (i.pl.), monitoring the nociceptive behavior, such as licking and flicking or shanking the injected paw, for 1 h and recording every 5 min, as previously reported [42].

The formalin injection induces a rapid pain response (phase I), which lasts about 5 min, due to direct nociceptors’ activation, characterized by an acute form of pain. After a short quiescent period, another behavioral pain form (phase II) occurs, considered more important clinically, characterized by an inflammatory component and persistent pain due to nociceptive sensitization in the dorsal horn of the spinal cord [43].

#### 3.6.5. Data Analysis

Results are expressed as mean ± S.E.M. (*n* = 8–10 per group). * *p* < 0.05, ** *p* < 0.01. Two-way ANOVA was followed by Bonferroni’s multiple comparison test. Statistical analyses were performed using GraphPad Prism version 9.3.1 (GraphPad Software, San Diego, CA, USA).

## 4. Conclusions

(−)-2*R*/*S*-LP2 (**1**), (−)-2*R*-LP2 (**2**), and (−)-2*S*-LP2 (**3**) compounds exhibited a multitarget opioid/σ1R profile. Instead, all (+)-2*R*/*S*-LP2 (**7**), (+)-2*R*-LP2 (**8**), and (+)-2*S*-LP2 (**9**) compounds displayed a profile of selective σ1R ligands. However, levo and desxtro isomers inhibited inflammatory pain induced in the mouse formalin test with a different mechanism. Thus, the multitarget or single target profile of *N*-normetazocine derivatives assayed could represent a promising pharmacological strategy to enhance opioid potency and/or increase the safety margin.

## 5. Future Directions

Using in vitro and in vivo profiles of (+)-2*R*/*S*-LP2 (**7**), we are investigating its therapeutic potential in neuropathic pain, specifically in a rodent model of unilateral sciatic nerve chronic constriction injury (CCI). Nowadays, neuropathic pain represents a significant burden for patients due to long-term therapeutic regimens and severe side effects. Moreover, a mechanistic study on σ1R involvement in the maintenance of painful states is ongoing.

## Data Availability

The data presented in this study are available on request from the corresponding author.

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
