# Peer review of "Novel N-normetazocine Derivatives with Opioid Agonist/Sigma-1 Receptor Antagonist Profile as Potential Analgesics in Inflammatory Pain"

_molecules, 2022, doi:10.3390/molecules27165135_

Round 1

Reviewer 1 Report

This manuscript clearly demonstrated the effect of (+)-2R/S-LP2 (7), (+)-2R-LP2 (8), (+)-2S-LP2 (9) compounds synthesized as sigma1 receptor selective antagonist on the formalin-induced inflammatory pain. The authors revealed that compound (7) but not (8) and (9) showed significant anti-inflammatory pain activity by intraperitoneal administration 20 min before the formalin test.

The results of this study are clear and it includes beneficial information to understand the effect of sigma1 receptor antagonist on the inflammatory pain, but I found several problems as following.

Specific comments

1. Authors had better show the data by using the isomer (+)-2S-LP2 (9) on the formalin test (Figure 4) even if the effect is similar to that of (8).

2. Although authors show the result of co-treatment of PRE-084 and (+)-2R/S-LP2, it is still speculative that (+)-2R/S-LP2 surely antagonize the sigma1 receptor in vivo, since the sigma1 agonist per se can potentiate the inflammatory pain by formalin via sigma1 receptor. In other words, it is clearer if the authors can determine the inhibitory effect of sigma-1 antagonist on sigma-1 agonist-induced pain, if it is possible. 

3. Authors had better add the information about the drug (i.e., LP compounds) administration in vivo in the materials and methods section. 

4. Authors had better to add some description on the possible endogenous sigma1 agonist on the inflammatory pain if any in the introduction section.

Author Response

This manuscript clearly demonstrated the effect of (+)-2R/S-LP2 (7), (+)-2R-LP2 (8), (+)-2S-LP2 (9) compounds synthesized as sigma1 receptor selective antagonist on the formalin-induced inflammatory pain. The authors revealed that compound (7) but not (8) and (9) showed significant anti-inflammatory pain activity by intraperitoneal administration 20 min before the formalin test.

The results of this study are clear and it includes beneficial information to understand the effect of sigma1 receptor antagonist on the inflammatory pain, but I found several problems as following.

We thank this reviewer for her/his time, courtesy, and expert review of our manuscript. We are glad that she/he found our manuscript of interest in study of sigma 1 receptor antagonist profile on inflammatory pain.

Specific comments

  1. Authors had better show the data by using the isomer (+)-2S-LP2 (9) on the formalin test (Figure 4) even if the effect is similar to that of (8).

R1. We thank the reviewer for his/her suggestion. The (+)-2S-LP2 anti-inflammatory effect is now attached.

  1. Although authors show the result of co-treatment of PRE-084 and (+)-2R/S-LP2, it is still speculative that (+)-2R/S-LP2 surely antagonize the sigma1 receptor in vivo, since the sigma1 agonist per se can potentiate the inflammatory pain by formalin via sigma1 receptor. In other words, it is clearer if the authors can determine the inhibitory effect of sigma-1 antagonist on sigma-1 agonist-induced pain, if it is possible. 

R2. We thank reviewer for his/her suggestion that increase the quality of the presentation of our evidence. We now provide a revised figure 5 with data on PRE-084 alone. In our animal model, the administration of sigma-1 receptor agonist alone didn’t show a significant change in nociceptive behavior, according to literature data (Turnaturi R, Pasquinucci L, Chiechio S, Grasso M, Marrazzo A, Amata E, Dichiara M, Prezzavento O, Parenti C. Exploiting the Power of Stereochemistry in Drug Action: 3-[(2S,6S,11S)-8-Hydroxy-6,11-dimethyl-1,4,5,6-tetrahydro-2,6-methano-3-benzazocin-3(2H)-yl]-N-phenylpropanamide as Potent Sigma-1 Receptor Antagonist. ACS Chem Neurosci. 2020 Apr 1;11(7):999-1005. doi: 10.1021/acschemneuro.9b00688. Epub 2020 Mar 23. PMID: 32186844), (Cobos EJ, Entrena JM, Nieto FR, Cendán CM, Del Pozo E. Pharmacology and therapeutic potential of sigma(1) receptor ligands. Curr Neuropharmacol. 2008 Dec;6(4):344-66. doi: 10.2174/157015908787386113. PMID: 19587856; PMCID: PMC2701284). but, as shown in Fig. 5, prevented (+)-2R/S-LP2 antinociceptive effect.

  1. Authors had better add the information about the drug (i.e., LP compounds) administration in vivo in the materials and methods section. 

R3. We now included this information in the materials and methods section. 

  1. Authors had better to add some description on the possible endogenous sigma1 agonist on the inflammatory pain if any in the introduction section.

R4. The introduction section was improved with data on endogenous ligands.

Reviewer 2 Report

In the present paper, the authors' work aimed to evaluate the opioid and/or σ1R functional profile of N-normetazocine-based ligands as potential therapeutic agents for inflammatory pain management. The pivotal role of N-normetazocine stereochemistry, by comparing the pharmacological activity of three newly synthesized isomers the (+)-2R/S-LP2 (7), (+)-2R-LP2 (8), and (+)-2S-LP2 (9) with three isomers the (-)-2R/S-LP2 (1), (-)-2R-LP2 (2) and (-)-2S-LP2 (3) synthesized previously, are discussed. The structures of all newly synthesized compounds were confirmed with spectroscopic methods (1H, 13C NMR, and elemental analysis). The resolution of (±)-cis-N-normetazocine and the synthesis of intermediates were carried out. For all (−)-LP2 and (+)-LP2 isomers the affinity for opioid- and σ1-, σ2-receptors were evaluated by competition binding assays. Moreover, all isomers were evaluated to the In Vivo formalin test. Reference compounds were included, representing usual control experiments.

 Taking into account the amount of work pharmacology is stronger than chemistry.

 Here are some points to be considered before publication:

 This was generally well written. I did note several instances where an English article [a, an, or the] was left out – it should be corrected,

 1. Introduction

Figure 1. (-)-LP2 isomers structures –compound’s code - (-)-LP2 (1) - should be replaced by (-)-2R/S-LP2 (1)

 2.1 Chemistry

 Scheme 1 – should be rewritten – the full structure of (+)-cis-N-normetazocine should be added instead of  “b”.

 3.2. General procedure for the synthesis of compounds 79

-  pages 7-8; All final compounds were purified by flash chromatography on a silica gel column using CH2Cl2/C2H5OH (97:3 v/v) as eluent. Chemical purities of the compounds are checked by classical TLC – there is a lack of the Rf value for each of the final compounds - this should be added in the experimental part 3.2.

 And finally, the subsection “Future Directions” - should be added.

 In summarizing: the scientific content of the reviewed paper fits the Journal's topical scope and the results of the study are fairly interesting. The applied methods are present as sound and appropriate. The figures are relevant and adequate. The conclusions drawn are justified by experimental results. The literature references are properly selected.

Conclusion:

If the manuscript is substantially modified this text will be acceptable for publication

Author Response

In the present paper, the authors' work aimed to evaluate the opioid and/or σ1R functional profile of N-normetazocine-based ligands as potential therapeutic agents for inflammatory pain management. The pivotal role of N-normetazocine stereochemistry, by comparing the pharmacological activity of three newly synthesized isomers the (+)-2R/S-LP2 (7), (+)-2R-LP2 (8), and (+)-2S-LP2 (9) with three isomers the (-)-2R/S-LP2 (1), (-)-2R-LP2 (2) and (-)-2S-LP2 (3) synthesized previously, are discussed. The structures of all newly synthesized compounds were confirmed with spectroscopic methods (1H, 13C NMR, and elemental analysis). The resolution of (±)-cis-N-normetazocine and the synthesis of intermediates were carried out. For all (−)-LP2 and (+)-LP2 isomers the affinity for opioid- and σ1-, σ2-receptors were evaluated by competition binding assays. Moreover, all isomers were evaluated to the In Vivo formalin test. Reference compounds were included, representing usual control experiments.

 Taking into account the amount of work pharmacology is stronger than chemistry.

 In summarizing: the scientific content of the reviewed paper fits the Journal's topical scope and the results of the study are fairly interesting. The applied methods are present as sound and appropriate. The figures are relevant and adequate. The conclusions drawn are justified by experimental results. The literature references are properly selected.

 We thank this reviewer for her/his time, courtesy, and expert review of our manuscript. We are glad that she/he found our manuscript of interest in study of sigma 1 receptor antagonist profile on inflammatory pain.

Here are some points to be considered before publication:

This was generally well written. I did note several instances where an English article [a, an, or the] was left out – it should be corrected.

R1. We re-checked all article in the manuscript.

  1. Introduction 

Figure 1. (-)-LP2 isomers structures –compound’s code - (-)-LP2 (1) - should be replaced by (-)-2R/S-LP2 (1)

R2. We thank the reviewer for this suggestion. We modified it accordingly.

 2.1 Chemistry

 Scheme 1 – should be rewritten – the full structure of (+)-cis-N-normetazocine should be added instead of  “b”.

R2. We thank the reviewer for this suggestion. We modified it accordingly.

3.2. General procedure for the synthesis of compounds 7−9  -  pages 7-8;

 All final compounds were purified by flash chromatography on a silica gel column using CH2Cl2/C2H5OH (97:3 v/v) as eluent. Chemical purities of the compounds are checked by classical TLC – there is a lack of the Rf value for each of the final compounds - this should be added in the experimental part 3.2.

R3. For each final compounds the Rf value was added in the experimental section.

And finally, the subsection “Future Directions” - should be added.

R4. We included this section in the revised version of the manuscript.

Reviewer 3 Report

After carefully reading the manuscript, I conclude that the summary, introduction and the remaining chapters provide an exhaustive and appropriate discussion of the issues raised.

Although it is a valuable work with an interesting idea, the chapter of results and discussion call for corrections:

          As the name of the journal suggests, it is a journal dedicated to molecules. Therefore, it seems necessary to supplement this manuscript with the information about structure of all analyzed compounds. The molecules selected for comparison: haloperidol, (+) - pentazocine, DTG, BD-1063, DAMGO, naltrindole and U69,593 have different structures compared to the tested systems (1-3, 7-9) (Table 1). As it is known, different chemical structures show different biological effects. In my opinion, this aspect requires commentary in the paper. In addition, I strongly encourage the authors to perform a simple similarity test by calculating the Tanimoto structural similarity index (Tsc) using the online software https://chemminetools.ucr.edu/similarity/. Please tabulate the obtained data and discuss them substantively.

          the discussion of the obtained results (Table 1) of efficacy should be based on the values of the ratio of inhibition of σ1R / σ2R receptors (and MOR / DOR or MOR / KOR, respectively) for each of the tested compounds.

          lines no. 185-186 contains a repetition of data gathered in Table 1, which is not a good practice in scientific paper. I recommend that at this point the authors also use the ratio Ki (+) - 2R / S-LP2 / (+) - 2R-LP2 and (+) - 2R / S-LP2 / (+) - 2S-LP2 and (+) -2R-LP2 / (+) - 2S-LP2.

 After clarifying these points and appropriate manuscript corrections, the work may be reconsidered for publication.

 I recommend publication after major revision.

Author Response

After carefully reading the manuscript, I conclude that the summary, introduction and the remaining chapters provide an exhaustive and appropriate discussion of the issues raised.

 We thank this reviewer for her/his time, courtesy, and expert review of our manuscript.

Although it is a valuable work with an interesting idea, the chapter of results and discussion call for corrections:

  • As the name of the journal suggests, it is a journal dedicated to molecules. Therefore, it seems necessary to supplement this manuscript with the information about structure of all analyzed compounds. The molecules selected for comparison: haloperidol, (+) - pentazocine, DTG, BD-1063, DAMGO, naltrindole and U69,593 have different structures compared to the tested systems (1-3, 7-9) (Table 1). As it is known, different chemical structures show different biological effects. In my opinion, this aspect requires commentary in the paper. In addition, I strongly encourage the authors to perform a simple similarity test by calculating the Tanimoto structural similarity index (Tsc) using the online software https://chemminetools.ucr.edu/similarity/. Please tabulate the obtained data and discuss them substantively.

R1. In our investigation Haloperidol, (+) - Pentazocine, DTG, BD-1063, DAMGO, Naltrindole and U69,593 were used as internal controls. We tested them with the same membrane homogenates used to evaluate the binding of our 1-3 and 7-9 compounds. Indeed, it is a good practice to report the affinity profile of well-established and well-known reference compounds for each receptor investigated through competition binding experiments. DTG (σ2R), (+) – Pentazocine (σ1R), BD-1063 (σ1R), DAMGO (MOR), Naltrindole (DOR) and U69,593 (KOR) showed Ki values for σ1R and σ2R and the three opioid receptors comparable to those reported in previous studies and in literature. Nevertheless, a similarity test by calculating the Tanimoto structural similarity index (Tsc) were conducted on all tested compounds.

We agree with reviewer about the sentence “different chemical structures show different biological effects”. Synthesized compounds 1-3 possess all pharmacophoric elements crucial for opioid receptor interaction such as the basic nitrogen and the 8-OH group that, in the rigid N-normetazocine scaffold, are at a distance comparable to that of endogenous opioid peptides. Moreover, the (-)-(2R,6R,11R) configuration of (-)-cis-N-normetazocine is identical to that of morphine.

Synthesized 7-9 compounds are their (+)-(2S,6S,11S) antipodes. In our paper we investigated the influence of N-normetazocine stereochemistry change (from levo to dextro configuration) in the efficacy profile of the dual-acting MOR/DOR agonist (-)-2R/S-LP2. Pioneering investigations, though the resolution of N-normetazocine-based compounds into the (-)- and (+)-isomers, established that the (+)-isomers are characterized by significant σ1R affinity, differently from the (-)-isomers that bind stronger to MOR and KOR. Thus, synthesized and in vitro and in vivo tested isomers (1-3 and 7-9), despite share the identical molecular formulas, atom-to-atom linkages, and bonding distances, are different chemical compounds. Indeed, in our investigation we demonstrated that the isomers differ in pharmacological properties because of the stereoselective isomer-target interaction that reflected different affinity, selectivity, and activity generating differences between isomers.

  • the discussion of the obtained results (Table 1) of efficacy should be based on the values of the ratio of inhibition of σ1R / σ2R receptors (and MOR / DOR or MOR / KOR, respectively) for each of the tested compounds.

R2. Thank you for the suggestion. We introduced the selectivity ratios in the discussion of the radioligand binding results.

  • lines no. 185-186 contains a repetition of data gathered in Table 1, which is not a good practice in scientific paper. I recommend that at this point the authors also use the ratio Ki (+) - 2R / S-LP2 / (+) - 2R-LP2 and (+) - 2R / S-LP2 / (+) - 2S-LP2 and (+) -2R-LP2 / (+) - 2S-LP2.

R3. Thank you for the observation. Now, we corrected the sentences avoiding the repetition of data gathered in Table 1 and we also introduced selectivity ratios.

Round 2

Reviewer 1 Report

Authors appropriately responded or answered to my comments and I think the revised manuscript is now suitable for publication in the Molecules.

Reviewer 3 Report

Thank you to the authors for addressing my comments. I believe the revised manuscript improved in clarity.